# Research on Intelligent Vehicle Trajectory Tracking Control Based on Improved Adaptive MPC

**DOI:** 10.3390/s24072316

**Published:** 2024-04-05

**Authors:** Wei Tan, Mengfei Wang, Ke Ma

**Affiliations:** Key Laboratory of Advanced Manufacturing Technology for Automobile Parts, Ministry of Education, Chongqing University of Technology, Chongqing 400054, China; wtan@cqut.edu.cn (W.T.); 17888575551@stu.cqut.edu.cn (M.W.)

**Keywords:** trajectory tracking, adaptive model predictive control (AMPC), unscented Kalman filter, adaptive modified estimation of tire cornering stiffness, tire lateral force estimation, dynamic prediction time domain

## Abstract

Intelligent vehicle trajectory tracking exhibits problems such as low adaptability, low tracking accuracy, and poor robustness in complex driving environments with uncertain road conditions. Therefore, an improved method of adaptive model predictive control (AMPC) for trajectory tracking was designed in this study to increase the corresponding tracking accuracy and driving stability of intelligent vehicles under uncertain and complex working conditions. First, based on the unscented Kalman filter, longitudinal speed, yaw speed, and lateral acceleration were considered as the observed variables of the measurement equation to estimate the lateral force of the front and rear tires accurately in real time. Subsequently, an adaptive correction estimation strategy for tire cornering stiffness was designed, an AMPC method was established, and a dynamic prediction time-domain adaptive model was constructed for optimization according to vehicle speed and road adhesion conditions. The improved AMPC method for trajectory tracking was then realized. Finally, the control effectiveness and trajectory tracking accuracy of the proposed AMPC technique were verified via co-simulation using CarSim and MATLAB/Simulink. From the results, a low lateral position error and heading angle error in trajectory tracking were obtained under different vehicle driving conditions and road adhesion conditions, producing high trajectory-tracking control accuracy. Thus, this work provides an important reference for improving the adaptability, robustness, and optimization of intelligent vehicle tracking control systems.

## 1. Introduction

Intelligent driving trajectory tracking control is the core technology for intelligent vehicle planning and control and provides the basis for intelligent vehicle automatic control [1]. However, regarding the strong nonlinear and time-varying characteristics in vehicle path tracking systems, the dynamic performance of the control algorithm must be sufficiently high. Accordingly, improvements to the tracking accuracy and robustness of controllers are urgently required.

Commonly used trajectory-tracking control algorithms include proportional–integral–derivative (PID) control [2], preview control [3,4,5], optimal control [6,7,8], and model predictive control (MPC) [9,10,11,12]. MPC can consider various safety factors and has thus attracted widespread attention from scholars for the optimal control of linear/nonlinear systems under physical constraints. Yakub et al. [13] compared the linear quadratic regulator with linear time-varying (LTV) MPC, and LTV-MPC demonstrated a better path tracking effect. Jianwei et al. [14] elaborated on the application of MPC in autonomous driving. Falcone et al. [15] proposed an LTV-MPC that can ensure the stability of a vehicle at high speeds through limiting the tire slip angle. Katriniok et al. [16] expanded upon the work in [15] and linearized the model within the prediction time, resulting in a high accuracy and control effect for the LTV-MPC. To increase the stability and safety of a vehicle when tracking the reference trajectory, Cao et al. [17] expanded the constraint conditions of the LTV-MPC to the center-of-mass sideslip angle, lateral acceleration, and tire sideslip angle. The obtained LTV-MPC exhibited strong adaptability. Li et al. [18] integrated the driver’s driving intention, driving environment assessment, and the MPC algorithm to form a shared fuzzy steering controller that can successfully assist drivers in avoiding obstacles and ensure vehicle stability. Zhong Siqi [19] proposed an MPC method for path-tracking lateral control (KNMPC) with an approximate tracking error to avoid duplication problems caused by finding the nearest point during operation, thus reducing errors and improving the operation efficiency.

However, none of the aforementioned studies included the impact of model errors on the MPC effect. Guo et al. [20] considered the model error as the measurement disturbance of the yaw angular velocity and proposed an MPC method based on a differential evolution algorithm that can improve the trajectory tracking accuracy of unmanned vehicles when determining the road adhesion coefficient. MPC requires the discretization and linearization of system dynamics, which leads to a range of modeling errors. Therefore, [21] designed an adaptive sliding mode controller for use in vehicle formations based on nonholonomic wheeled mobile robots. This controller has the advantage of dealing with the unknown complex behavior of friction, the uncertainty of parameters, and external interference without prior knowledge of parameters and structures. The simulation results show that the performance of this controller represents a significant improvement compared with other existing controllers. Reference [22] designed a path tracking controller based on robust output feedback that can be applied to a class of self-reconfiguring mobile robots under actuator saturation. The controller can estimate unmeasurable states and uncertain dynamic items, a fast dynamic compensator is used to meet the actuator constraints and limitations, and the closed-loop stability of the system is analyzed based on the contraction theory, which is better than Lyapunov’s. The simulation results show that the controller has better control performance. In addition, Xuanxuan proposed improved MPC based on fuzzy control and classic MPC. In particular, the weight value and cost function can be adjusted according to the lateral position and route errors, thereby improving the tracking accuracy and comfort of the vehicle during the continuous tracking process [23]. Xudong et al. [24] focused on the problem of the poor adaptability of MPC systems and proposed an emergency obstacle avoidance strategy based on trajectory replanning and path-tracking dual-layer MPC, fully utilizing the advantages of the algorithm. Zhenyu [25] designed an obstacle avoidance controller based on a BP neural network and MPC. With the simplified parameter adjustment process and optimization of weight parameters, this method exhibits high tracking accuracy. Shi et al. [26] built an adaptive MPC (AMPC) system that can operate under the condition of a disturbed path curvature, based on the kinematic model and dynamic driving deviation model of autonomous wheeled loaders. The results show that its tracking effect is significantly improved compared with that of traditional MPC. Wang et al. [27] designed a highly adaptive and robust MPC method that can respond to uncertain interference to address the slipping problem caused by vehicles driving on rough and unstable roads.

However, owing to the influence of the road adhesion coefficient and vertical load, a complex nonlinear relationship exists between the tire force and slip angle [28]. Large tire slip angles cause nonlinear tire slip characteristics, resulting in large model errors, as well as poor tracking accuracy and vehicle driving stability [29], leading to dangerous accidents [30]. Therefore, current MPC methods cannot satisfy the tracking requirements under complex road conditions based on linear constant tire cornering stiffness. In addition, the predicted time domain in general MPC systems is often considered a fixed value. However, the choice of the predicted time domain significantly impacts the control effect of the system. Differences in the predicted time domains for various driving speeds and road adhesion conditions lead to different control effects. It is thus necessary to examine the impact of different predicted time domains on the control effect under different driving speeds and road adhesion conditions.

Accordingly, this study proposes an improved AMPC method that considers the tire lateral force calculation deviation and dynamically predicts a time-domain adaptive model for different driving states and road adhesion conditions of a vehicle. First, the unscented Kalman filter (UKF) algorithm is employed based on a simplified four-wheel vehicle body dynamics model and variables such as longitudinal vehicle speed, yaw angular velocity, and lateral acceleration are considered as the observed quantities of the measurement equation to accurately estimate the lateral forces of the front and rear tires in real time. Based on this, an adaptive correction estimation strategy for tire cornering stiffness is designed. The front- and rear-tire lateral forces (${\hat{F}}_{yf}$ and ${\hat{F}}_{yr}$, respectively), accurately estimated in real time using the UKF, and the corresponding linear tire lateral forces ($F_{yf}$ and $F_{yr}$), based on the linear constant tire cornering stiffness, are used to perform comparisons and other operations. The front- and rear-tire cornering stiffness correction factors are then defined, and the AMPC is established. A dynamic prediction time-domain adaptive model is designed based on vehicle speed and road adhesion conditions to further improve the proposed AMPC, and an improved AMPC method for trajectory tracking is realized. Finally, the control effectiveness and trajectory tracking accuracy of the proposed improved AMPC method are verified through a joint simulation using CarSim 2019 and MATLAB/Simulink 2021b software.

The rest of this paper is organized as follows: Section 2 introduces the vehicle–road error dynamics model, including the four-wheel vehicle body dynamics model and the two-wheel autonomous vehicle dynamics model. Section 3 presents estimations of the tire lateral force and tire cornering stiffness based on the established dynamics model described in Section 2 and outlines the verification of the simulation results. In Section 4, the design of the improved AMPC method for trajectory tracking is detailed. Section 5 provides the simulation test results and verification of the proposed tracking controller. Finally, the study is summarized in Section 6.

## 2. Vehicle–Road Error Dynamics Model

### 2.1. Vehicle Dynamics Model

The tire lateral force is estimated based on the vehicle dynamics model, which completes the estimation of the tire cornering stiffness. Therefore, a four-wheel vehicle body model (also known as a dual-track model) was established to effectively simulate a vehicle’s motion characteristics on flat roads considering the force conditions of the vehicle’s four tires.

In Figure 1, m is the mass of the vehicle; $I_{z}$ is the moment of inertia of the vehicle around the *z*-axis; $F_{xfl},\;F_{xfr},\;F_{xrl}$, and $F_{xrr}$ are the longitudinal forces of the left front, right front, left rear, and right rear tires, respectively; $F_{yfl},\;F_{yfr},\;F_{yrl}$, and $F_{yrr}$ correspond to the lateral forces of the left front, right front, left rear, and right rear tires; $\alpha_{fl},\;\alpha_{fr},\;\alpha_{rl}$, and $\alpha_{rr}$ are the corresponding slip angles of the four tires; $l_{f}$ and $l_{r}$ represent the distances between the center of mass and the front and rear axles, respectively; $v_{x},\;v_{y}$, and v are the longitudinal, transverse, and combined speeds at the center of mass of the vehicle; $W_{d}$ is the left and right wheelbase; $\delta_{f}$ is the front-wheel rotation angle; and $\dot{\varphi }$ is the yaw angular velocity. The longitudinal, transverse, and yaw dynamic equations of the four-wheel body model are expressed in Equation (1).
(1)$$\left\{\begin{array}{l} ma_{x}=F_{xrl}+F_{xrr}+F_{xfl}\cos\delta_{f}+F_{xfr}\cos\delta_{f}-F_{yfl}\sin\delta_{f}\\ \;\;\;-F_{yfr}\sin\delta_{f}\\ ma_{y}=F_{yrl}+F_{yrr}+F_{yfl}\cos\delta_{f}+F_{yfr}\cos\delta_{f}+F_{xfl}\sin\delta_{f}\\ \;\;\;+F_{xfr}\sin\delta_{f}\\ I_{z}\ddot{\varphi }=l_{f}(F_{yfl}\cos\delta_{f}+F_{yfr}\cos\delta_{f}+F_{xfl}\sin\delta_{f})\\ \;\;\;+\frac{W_{d}}{2}(F_{yfl}\sin\delta_{f}-F_{yfr}\sin\delta_{f}-F_{xfl}+F_{xfr}-F_{xrl}\\ \;\;\;+F_{xrr})-l_{r}(F_{yrl}+F_{yrr})\\ a_{x}={\dot{v}}_{x}-v_{y}\dot{\varphi }\\ a_{y}={\dot{v}}_{y}-v_{x}\dot{\varphi } \end{array}\right.$$

Meanwhile, active steering control of the front wheels is used to realize the lateral path-tracking control of the smart car. Taking the front wheel angle as the input and ignoring the role of the suspension, the car has only two degrees of freedom (DOFs): lateral motion along the *y*-axis and yaw motion around the *z*-axis. The established 2-DOF dynamics model is illustrated in Figure 2.

In Figure 2, $F_{yf}$ and $F_{yr}$ are the lateral forces on the front and rear wheels, respectively; $\alpha_{f}$ and $\alpha_{r}$ are the sideslip angles of the front and rear wheels, respectively; $\delta_{f}$ is the front-wheel rotation angle; β is the center-of-mass sideslip angle; $v_{x}$ and $v_{y}$ correspond to the longitudinal velocity and the lateral velocity at the center of mass of the vehicle; m is the mass of the vehicle; $I_{z}$ is the moment of inertia of the vehicle around the *z*-axis; $\dot{\varphi }$ is the yaw angular velocity; and $l_{f}$ and $l_{r}$ are the distances between the center of mass and the front and rear axles, respectively. The dynamic equations for the lateral and yaw motions of the vehicle are as follows:(2)$$F_{yf}\cos(\delta_{f})+F_{yr}=m(v_{y}+v_{x}\dot{\varphi })$$
(3)$$l_{f}F_{yf}\cos(\delta_{f})-l_{r}F_{yr}=I_{z}\ddot{\varphi }$$

Under the small-angle assumption, the front- and rear-wheel slip angles can be expressed as follows:(4)$$\left\{\begin{array}{l} \alpha_{f}=\arctan\left(\frac{l_{f}\dot{\varphi }+v_{y}}{v_{x}}\right)-\delta_{f}\approx \frac{l_{f}\dot{\varphi }+v_{y}}{v_{x}}-\delta_{f}\\ \alpha_{r}=\arctan\left(\frac{v_{y}-l_{r}\dot{\varphi }}{v_{x}}\right)\approx \frac{v_{y}-l_{r}\dot{\varphi }}{v_{x}} \end{array}\right.$$

The lateral forces on the front and rear tires for smaller tire slip angles can be described by a linear relationship [31]. Combined with Equation (3), the lateral forces $F_{yf}$ and $F_{yr}$ on the front and rear wheels can be obtained as follows:(5)$$\left\{\begin{array}{l} F_{yf}=C_{\alpha f}\alpha_{f}=C_{\alpha f}\left(\frac{l_{f}\dot{\varphi }+v_{y}}{v_{x}}-\delta_{f}\right)\\ F_{yr}=C_{\alpha r}\alpha_{r}=C_{\alpha r}\left(\frac{v_{y}-l_{r}\dot{\varphi }}{v_{x}}\right) \end{array}\right.$$
where $C_{\alpha f}$ and $C_{\alpha r}$ indicate the total cornering stiffness of the tires on both sides of the vehicle’s front and rear axles, respectively, and $\alpha_{f}$ and $\alpha_{r}$ correspond to the front- and rear-tire sideslip angles.

Combining Equations (2), (3) and (5), the simplified form of the vehicle’s 2-DOF dynamic equation can be obtained as follows:(6)$$\left\{\begin{array}{l} {\dot{v}}_{y}=\left(\frac{l_{f}C_{\alpha f}-l_{r}C_{\alpha r}}{mv_{x}}-v_{x}\right)\dot{\varphi }+\left(\frac{C_{\alpha f}+C_{\alpha r}}{mv_{x}}\right)v_{y}-\frac{C_{\alpha f}}{m}\delta_{f}\\ \ddot{\varphi }=\frac{l_{f}C_{\alpha f}}{I_{z}}(\frac{l_{f}\dot{\varphi }+v_{y}}{v_{x}}-\delta_{f})-\frac{l_{r}C_{\alpha r}}{I_{z}}(\frac{v_{y}-l_{r}\dot{\varphi }}{v_{x}}) \end{array}\right.$$

### 2.2. Vehicle–Road Error Dynamics Model

To express the relationship between the vehicle and the reference trajectory, a trajectory-tracking vehicle–road error model was established, as depicted in Figure 3. The inertial coordinate system, XOY, is fixed to Earth. The position of the vehicle’s center of mass in the figure is marked as point *O*.

In Figure 3, φ is the vehicle’s yaw angle, $e_{d}$ is the distance between the vehicle’s center of mass and the closest point (c) on the desired trajectory, p is the preview point, L is the preview distance, and $e_{dp}$ is the lateral direction from the vehicle’s center of mass to the preview point (p) on the desired trajectory. The vertical distance ($\varphi_{p}$) is the desired heading angle at the preview point.

The difference between the vehicle’s yaw angle and the expected heading angle ($\theta_{r}$) from the nearest reference point on the path is expressed as the heading angle deviation, recorded as $e_{\varphi }$; then, ${\dot{e}}_{\varphi }$ and ${\ddot{e}}_{\varphi }$ can be expressed as follows:(7)$$\left\{\begin{array}{c} {\dot{e}}_{\varphi }=\dot{\varphi }-{\dot{\theta }}_{r}\\ {\ddot{e}}_{\varphi }=\ddot{\varphi }-{\ddot{\theta }}_{r} \end{array}\right.$$

When the vehicle travels normally along the road with curvature ρ at the reference point, ${\dot{\theta }}_{r}$ can be expressed as follows:(8)$${\dot{\theta }}_{r}=\rho v_{x}$$

Under normal circumstances, road curves are relatively gentle and the relative curvature changes more slowly, so ${\ddot{\theta }}_{r}$ can be ignored.

Similarly, ${\dot{e}}_{d}$ and ${\ddot{e}}_{d}$ are given by the following equations:(9)$$\left\{\begin{array}{c} {\dot{e}}_{d}=v_{x}e_{\varphi }+v_{y}\\ {\ddot{e}}_{d}=v_{x}{\dot{e}}_{\varphi }+{\dot{v}}_{y} \end{array}\right.$$

Therefore, the complete path-tracking error dynamics model can be obtained by combining Equations (6)–(9):(10)$$\dot{X}=A_{1}X+B_{1}u+C_{1}{\dot{\theta }}_{r}$$

Matrices X, $A_{1}$, $B_{1}$, $C_{1}$, and *u* are defined as follows:$$X=\left[\begin{array}{c} e_{d}\\ {\dot{e}}_{d}\\ e_{\varphi }\\ {\dot{e}}_{\varphi } \end{array}\right],\;u=\delta_{f},\;A_{1}=\left[\begin{array}{cccc} 0 & 1 & 0 & 0\\ 0 & \frac{C_{\alpha f}+C_{\alpha r}}{mv_{x}} & -\frac{C_{\alpha f}+C_{\alpha r}}{m} & \frac{l_{f}C_{\alpha f}-l_{r}C_{\alpha r}}{mv_{x}}\\ 0 & 0 & 0 & 1\\ 0 & \frac{l_{f}C_{\alpha f}-l_{r}C_{\alpha r}}{l_{z}v_{x}} & -\frac{l_{f}C_{\alpha f}-l_{r}C_{\alpha r}}{l_{z}} & \frac{{l_{f}}^{2}C_{\alpha f}+{l_{r}}^{2}C_{\alpha r}}{l_{z}v_{x}} \end{array}\right],\;\;B_{1}=\left[\begin{array}{c} 0\\ \frac{-C_{\alpha f}}{m}\\ 0\\ \frac{-l_{f}C_{\alpha f}}{I_{z}} \end{array}\right],\;C_{1}=\left[\begin{array}{c} 0\\ \frac{l_{f}C_{\alpha f}-l_{r}C_{\alpha r}}{mv_{x}}-v_{x}\\ 0\\ \frac{l_{f}^{2}C_{\alpha f}+l_{r}^{2}C_{\alpha r}}{l_{z}v_{x}} \end{array}\right].$$

## 3. Tire Lateral Force and Cornering Stiffness Estimation

### 3.1. Tire Lateral Force Estimation Based on Unscented Kalman Filter (UKF)

Owing to the vehicle tire load transfer and road adhesion changes, the tire cornering stiffness often varies. Therefore, using a constant tire cornering stiffness as the prediction model for the lateral MPC results in inaccuracies and affects the control effectiveness and tracking accuracy of the lateral path-tracking controller. Therefore, it is necessary to accurately estimate the tire lateral force in real time and further combine the estimated tire lateral force to perform a real-time correction estimation of the tire cornering stiffness.

Accordingly, the four-wheel vehicle body dynamics model mentioned in Section 2.1, which reflects the actual dynamic characteristics of a vehicle, was employed in this work to design a nonlinear observer. In addition, UKF was used to estimate tire lateral force.

#### 3.1.1. Design of Tire Lateral Force Estimator Based on UKF

The 3-DOF dynamics model of a four-wheel vehicle body does not include prior knowledge of the tire force model and road friction; therefore, the random walk model can be used to describe changes in various lateral and longitudinal forces [32]:(11)$$\left[\begin{array}{l} {\dot{F}}_{xfl}\\ {\dot{F}}_{xfr}\\ {\dot{F}}_{yfl}\\ {\dot{F}}_{yfr}\\ {\dot{F}}_{yrl}\\ {\dot{F}}_{yrr} \end{array}\right]=\left[\begin{array}{l} 0\\ 0\\ 0\\ 0\\ 0\\ 0 \end{array}\right]$$

Averaging the left and right-tire sideslip angle models, the front- and rear-tire sideslip angles are obtained as follows:(12)$$\left\{\begin{array}{l} \alpha_{f}=\alpha_{fl}=\alpha_{fr}=\arctan\left(\frac{l_{f}\dot{\varphi }+v_{y}}{v_{x}}\right)-\delta_{f}\\ \alpha_{r}=\alpha_{rl}=\alpha_{rr}=\arctan\left(\frac{v_{y}-l_{r}\dot{\phi }}{v_{x}}\right) \end{array}\right.$$

To ensure the observability of the system further, the differences between the left and right tires were assumed to be proportional to the changes in the tire load distribution.
(13)$$\left\{\begin{array}{c} F_{xfl}=\frac{F_{zfl}}{F_{zfl}+F_{zfr}}F_{xf},\;F_{xfr}=\frac{F_{zfr}}{F_{zfl}+F_{zfr}}F_{xf}\\ F_{yfl}=\frac{F_{zfl}}{F_{zfl}+F_{zfr}}F_{yf},\;F_{yfr}=\frac{F_{zfr}}{F_{zfl}+F_{zfr}}F_{yf}\\ F_{yrl}=\frac{F_{zrl}}{F_{zrl}+F_{zrr}}F_{yr},\;F_{yrr}=\frac{F_{zrr}}{F_{zrl}+F_{zrr}}F_{yr} \end{array}\right.$$
where $F_{zij}$ is the vertical force on each tire; $F_{yf}$ denotes the sum of the lateral forces of the front wheels, $F_{yf}=F_{yfl}+F_{yfr}$; $F_{yr}$ denotes the sum of the lateral forces of the rear wheels, $F_{yr}=F_{yrl}+F_{yrr}$; and $F_{xf}$ is the sum of the longitudinal forces of the front wheels, $F_{xf}=F_{xfl}+F_{xfr}$. In addition, the calculation method for the $F_{zij}$ vertical load is represented by a simple linear model [33]:(14)$$\left\{\begin{array}{c} F_{zfl}=\frac{l_{r}mg}{2L}-\frac{hma_{x}}{2L}-\frac{hl_{r}ma_{y}}{W_{d}L}\\ F_{zfr}=\frac{l_{r}mg}{2L}-\frac{hma_{x}}{2L}+\frac{hl_{r}ma_{y}}{W_{d}L}\\ F_{zrl}=\frac{l_{f}mg}{2L}+\frac{hma_{x}}{2L}-\frac{hl_{f}ma_{y}}{W_{d}L}\\ F_{zrr}=\frac{l_{f}mg}{2L}+\frac{hma_{x}}{2L}+\frac{hl_{f}ma_{y}}{W_{d}L} \end{array}\right.$$
where $W_{d}$ is the distance from the left wheels to the right wheels and h is the height of the vehicle’s center of mass. To sum up, let the state vector $X=[\dot{\varphi },v_{x},v_{y},F_{yf},F_{yr},F_{xf}]$ and the observation vector $Z=[\dot{\varphi },v_{x},a_{x},a_{y}]$. Combining the 3-DOF dynamics model of the four-wheel vehicle body mentioned in Section 2.1, Equations (1) and (11)–(14) are collapsed and discretized to obtain the state-space expression of the nonlinear system as follows:(15)$$\left\{\begin{array}{l} X(k)=f(X(k-1),U(k))+w(k)\\ Z(k)=h(X(k))+v(k) \end{array}\right.$$
where $U=\left[\begin{array}{ccccc} \delta_{f} & F_{zfl} & F_{zfr} & F_{zrl} & F_{zrr} \end{array}\right]$ is the control input vector; *X* is the state vector; Z is the observation vector; ω(k) and v(k) are the process noise and the observation noise, respectively, assumed to be Gaussian white noise; and f(⋅) and h(⋅) indicate the state transition equation and the observation equation, respectively. Let *X*, *Z*, and *U* be simplified as follows:(16)$$\left\{\begin{array}{l} X=\left[\begin{array}{cccccc} x_{1} & x_{2} & x_{3} & x_{4} & x_{5} & x_{6} \end{array}\right]\\ Z=\left[\begin{array}{cccc} z_{1} & z_{2} & z_{3} & z_{4} \end{array}\right]\\ U=\left[\begin{array}{ccccc} u_{1} & u_{2} & u_{3} & u_{4} & u_{5} \end{array}\right] \end{array}\right.$$

Therefore, the prediction equation of the nonlinear state function (f) and the observation function equation (h) can be obtained.
(17)$$f(k)=\left\{\begin{array}{l} f_{1}(k)=x_{1}(k-1)+\frac{T}{I_{z}}[l_{f}[x_{4}(k-1)\cos u_{1}(k)+x_{6}(k-1)\sin u_{1}(k)]-l_{r}x_{5}(k-1)+\frac{W_{d}}{2}[\frac{u_{2}(k)}{u_{2}(k)+u_{3}(k)}\cdot \\ \;\;\;x_{4}(k-1)\sin u_{1}(k)-\frac{u_{3}(k)}{u_{2}(k)+u_{3}(k)}x_{4}(k-1)\sin u_{1}(k)-\frac{u_{2}(k)}{u_{2}(k)+u_{3}(k)}x_{6}(k-1)\cos u_{1}(k)\\ \;\;\;+\frac{u_{3}(k)}{u_{2}(k)+u_{3}(k)}x_{6}(k-1)\cos u_{1}(k)]]\\ f_{2}(k)=x_{2}(k-1)+Tx_{1}(k-1)x_{3}(k-1)+\frac{T}{m}[x_{6}(k-1)\cos u_{1}(k)-x_{4}(k-1)\sin u_{1}(k)]\\ f_{3}(k)=x_{3}(k-1)-Tx_{1}(k-1)x_{2}(k-1)+\frac{T}{m}[x_{5}(k-1)+x_{4}(k-1)\cos u_{1}(k)+x_{6}(k-1)\sin u_{1}(k)]\\ f_{4}(k)=x_{4}(k-1)\\ f_{5}(k)=x_{5}(k-1)\\ f_{6}(k)=x_{6}(k-1) \end{array}\right.$$
where *T* denotes the discrete sampling time. Similarly, the observation function equation (h(⋅)) is given by the following formula:(18)$$h(k)=\left\{\begin{array}{l} h_{1}(k)=x_{1}(k)\\ h_{2}(k)=x_{2}(k)\\ h_{3}(k)=\frac{1}{m}[x_{6}(k)\cos u_{1}(k)-x_{4}(k)\sin u_{1}(k)]\\ h_{1}(k)=\frac{1}{m}[x_{5}(k)+x_{4}(k)\cos u_{1}(k)+x_{6}(k)\sin u_{1}(k)] \end{array}\right.$$

Combined with the formula above, the estimation algorithm based on the UKF proceeds as follows:

(1) System state initialization.

Assuming that the initial state (*X*(0)) is a random vector of a Gaussian distribution, the mean and variance of the initial state are as follows:(19)$$\left\{\begin{array}{l} {\hat{X}}_{0}=E(X_{0})\\ P_{0}=\mathrm{cov}(X_{0},{X_{0}}^{T})=E[(X_{0}-{\hat{X}}_{0}){\left(X_{0}-{\hat{X}}_{0}\right)}^{T}] \end{array}\right.$$

(2) Update and calculate the weighting coefficients.
(20)$$\left\{\begin{array}{l} {\omega_{0}}^{m}=\frac{\lambda }{n+\lambda }\\ {\omega_{0}}^{c}=\frac{\lambda }{n+\lambda }+(1-\alpha^{2}+\beta )\\ {\omega_{i}}^{c}={\omega_{i}}^{m}=\frac{\lambda }{2(n+\lambda )},\;i=1,\;2,\;3,\ldots ,\;2n \end{array}\right.$$
where $\lambda =n(\alpha^{2}-1)$ and β is a non-negative constant. Under normal circumstances, β is taken as 2 and α is the distribution state of the control sigma point, with a value of 0–1 (the value being taken as 0.2 in this article).

(3) Calculate and update the sigma points.
(21)$$X^{\left(i\right)}(k|k)=\left\{\begin{array}{l} \hat{X}(k|k),i=0\\ \hat{X}(k|k)+\sqrt{\left(n+\lambda )P(k|k\right)}\\ ,i=1,\ldots ,n\\ \hat{X}(k|k)-\sqrt{\left(n+\lambda )P(k|k\right)}\\ ,i=n+1,\ldots ,2n \end{array}\right.$$
where P is positive definite and $\sqrt{\left(n+\lambda )P(k|k\right)}$ is obtained through Cholesky decomposition.

(4) One-step prediction of system state quantities and covariance matrix calculation.

The predictions of a set of sigma points and their calculated weighted averages were used to obtain a one-step prediction of the system state quantity.
(22)$$\left\{\begin{array}{l} \hat{X}(k+1|k)=\sum_{i=0}^{2n}\omega^{\left(i\right)}X^{\left(i\right)}(k+1|k)\\ P(k+1|k)=\sum_{i=0}^{2n}\omega^{\left(i\right)}[\hat{X}(k+1|k)-X^{\left(i\right)}(k+1|k)]{\left[\hat{X}(k+1|k)-X^{\left(i\right)}(k+1|k)\right]}^{T}+Q \end{array}\right.$$

(5) Based on the one-step prediction value, the UT transform was used again to generate a new sigma point set.
(23)$$X^{\left(i\right)}(k+1|k)=\left\{\begin{array}{l} \hat{X}(k+1|k),\;i=0\\ \hat{X}(k+1|k)+\sqrt{\left(n+\lambda )P(k+1|k\right)}\;,i=1,\ldots ,n\\ \hat{X}(k+1|k)-\sqrt{\left(n+\lambda )P(k+1|k\right)}\;,i=n+1,\ldots ,2n \end{array}\right.$$

(6) The sigma point set predicted in step (5) was substituted into the observation equation to obtain a one-step prediction of the observation.
(24)$$Z^{\left(i\right)}(k+1|k)=h[X^{\left(i\right)}(k+1|k)]$$
where i=1,2,⋯,2n+1.

(7) The weighted sum of the observed and predicted values of the sigma point set obtained in the previous step was used to obtain the mean and covariance of the observed system predictions.
(25)$$\left\{\begin{array}{l} \bar{Z}(k+1|k)=\sum_{i=0}^{2n}\omega^{\left(i\right)}Z^{\left(i\right)}(k+1|k)\\ \;\;P_{z_{k}z_{k}}=\sum_{i=0}^{2n}\omega^{\left(i\right)}[Z^{\left(i\right)}(k+1|k)-\bar{Z}(k+1|k)][Z^{\left(i\right)}(k+1|k)-\bar{Z}(k+1|k){]}^{T}+R\\ \;\;P_{x_{k}z_{k}}=\sum_{i=0}^{2n}\omega^{\left(i\right)}[X^{\left(i\right)}(k+1|k)-\bar{X}(k+1|k)][Z^{\left(i\right)}(k+1|k)-\bar{Z}(k+1|k){]}^{T} \end{array}\right.$$

(8) Calculate the Kalman gain matrix *K*.
(26)$$K(k+1)=P_{x_{k}z_{k}}{P^{-1}}_{z_{k}z_{k}}$$

(9) Estimate the system state and update the covariance matrix (P).
(27)$$\left\{\begin{array}{l} \hat{X}(k+1)=\hat{X}(k+1|k)+K(k+1)[Z(k+1)-\hat{Z}(k+1|k)]\\ P(k+1)=P(k+1|k)-K(k+1)P_{z_{k}z_{k}}K^{T}(k+1) \end{array}\right.$$

#### 3.1.2. Simulation Verification of Tire Lateral Force Estimation

A joint simulation platform of CarSim and Simulink was built to conduct the simulation tests and verify the designed tire lateral force estimator. The parameters of the vehicle model considered in this simulation are listed in Table 1.

Working condition 1: The road adhesion coefficient was set to 0.9, and the vehicle was continuously turned left and right to observe the estimation accuracy, among other performances of the tire lateral force estimator during the changing steering process of the vehicle. The steering wheel input was a sinusoidal change with an amplitude of 180° and a period of 12.5 s; the vehicle speed changed as follows: within 0–10 s, the vehicle speed increased from 0 to 72 km/h, then maintained a period of constant speed. After 30 s, the vehicle decelerated to 54 km/h and finally continued driving at a constant speed [34].

The initial parameter settings of the UKF estimator include: the state quantity estimation initial value, ${\hat{X}}_{0}=[0;0;0;0;0;0]$; the error covariance matrix, $P_{0}=eye(6)$; the process noise covariance matrix; and the measurement noise covariance matrix, corresponding to Q=diag([0.05 0.01 0.01 226 127 1000]) and $R=diag([10^{-2};10^{-2};10^{-2};10^{-2}])$; and the sampling period of the system, estimated as 0.01 s.

The simulation results are presented in Figure 4.

In Figure 4, Figure 5, Figure 6, Figure 7 and Figure 8, the actual value of the tire lateral force is provided by the output of Carsim (that is, when Carsim and Simulink are jointly simulated, the tire lateral force output interface of the CarSim S-Function module in Simulink outputs the true value of the tire lateral force). In Figure 4, Figure 5, Figure 6, Figure 7 and Figure 8, the true value of the tire slip angle is also provided by the output of Carsim, which is compared with the estimated value for verification.

In the working condition simulation of tire lateral force estimation in Figure 4 and Figure 5, the estimated values of front- and rear-axle tire lateral force are based on the estimation output of the unscented Kalman filter (UKF) estimator to verify the effectiveness and accuracy of the tire lateral force estimation method based on UKF.

Under this relatively severe working condition, the simulation results in Figure 4a,b clarify the relatively small difference between the estimated vehicle tire lateral force and the real output of CarSim, in which the maximum errors of the estimated front- and rear-wheel lateral forces compared with the real values are approximately 687.9523 N and 386.4086 N, respectively, and the estimated lateral tire force matches well with the real value of the lateral tire force in the entire simulation process.

To confirm the accuracy of the tire lateral force estimator, working condition 2 was set, i.e., the double-lane-shift working condition was used as a scenario to verify the true reflection of the tire lateral force estimation under the complex working conditions of the emergency steering of the vehicle.

Working condition 2: The vehicle speed in the double-lane-shift condition was set to 72 km/h, and the road adhesion coefficient was set to 0.4. Figure 5 displays the simulation estimation results for the lateral forces on the front and rear tires.

From the simulation results in Figure 5a,b, even under complex working conditions with low adhesion, the estimated vehicle tire lateral force slightly differs from the actual output value of CarSim. The maximum errors of the front- and rear-wheel lateral force estimation values compared with the real values are approximately 634.7746 N and 670.4724 N, respectively. Moreover, the errors between the estimated lateral tire force and the real value are small throughout the simulation process, and the estimation accuracy is good.

The simulation results demonstrate that the designed estimator can accurately estimate the lateral force of a tire under complex working conditions.

### 3.2. Adaptive Correction Estimation of Tire Cornering Stiffness

The cornering stiffness was identified using the relationship between the lateral force of the tire and the tire slip angle, expressed as follows:(28)$$\left\{\begin{array}{l} F_{yf}=C_{f}\alpha_{f}\\ F_{yr}=C_{r}\alpha_{r} \end{array}\right.$$
for which the tire slip angle can be obtained using the following:(29)$$\left\{\begin{array}{l} \alpha_{f}=\arctan((l_{f}\dot{\varphi }+v_{y})/v_{x})-\delta_{f}\\ \alpha_{r}=\arctan((v_{y}-l_{r}\dot{\varphi })/v_{x}) \end{array}\right.$$

In the trajectory tracking control of intelligent driving vehicles, due to the influence of the road adhesion coefficient (μ) and the vertical load ($F_{z}$), there is a complex nonlinear relationship between tire force and slip angle. Under normal road conditions, when the wheel rotation angle is small, it can be assumed that the tire cornering stiffness remains unchanged [35]. When the road adhesion condition and vehicle driving states continue to change, a linear assumption is made regarding the cornering stiffness, and the calculated lateral tire force in the linear zone has a large error with respect to the real value, which affects the control performance of the tracking controller.

In this study, the tire cornering stiffness adaptive correction estimation strategy was adopted to complete the real-time correction estimation of the tire cornering stiffness as well as to eliminate the difference between the linear tire lateral force based on linear stiffness and the true value. The specific strategy involves defining the correction factors for the front- and rear-tire cornering stiffnesses via calculation of the difference between the front- and rear-tire lateral forces (${\hat{F}}_{yf}$ and ${\hat{F}}_{yr}$), accurately estimated in real time by the UKF, and the linear tire lateral forces ($F_{yf}$ and $F_{yr}$), based on the linear constant tire cornering stiffness.
(30)$$\left\{\begin{array}{c} \lambda_{f}=\frac{{\hat{F}}_{\mathrm{yf}}-F_{\mathrm{yf}}}{|{\hat{F}}_{\mathrm{yf}}|}\\ \lambda_{r}=\frac{{\hat{F}}_{\mathrm{yr}}-F_{\mathrm{yr}}}{|{\hat{F}}_{\mathrm{yr}}|} \end{array}\right.$$

Therefore, the corrected final values for the front- and rear-tire cornering stiffnesses can be expressed as follows:(31)$$\left\{\begin{array}{c} {\hat{C}}_{f}=(1+\lambda_{f})C_{f}\\ {\hat{C}}_{r}=(1+\lambda_{r})C_{r} \end{array}\right.$$

In addition, to avoid failure of the tire lateral force estimation caused by abnormal measurement noise and excessive change in the correction factor, which leads to deterioration of the controller’s stability, the correction factors for the front- and rear-tire cornering stiffnesses are constrained as follows:(32)$$\left\{\begin{array}{l} \lambda_{f\min}\leq \lambda_{f}\leq \lambda_{f\max}\\ \lambda_{r\min}\leq \lambda_{r}\leq \lambda_{r\max} \end{array}\right.$$

Finally, based on the simulation experiments and experience and other references, $\lambda_{f\;\min}=\lambda_{r\;\min}=-0.6$ and $\lambda_{f\;\max}=\lambda_{r\;\max}=1$. To avoid the singular value of the correction factor ($\lambda_{f}$ or $\lambda_{r}$) when the estimated value of the tire force is 0, for a tire side deflection angle ($\alpha_{f}$ or $\alpha_{r}$) of less than 0.2°, the corresponding correction factor will be considered as 0.

### 3.3. Simulation Verification of Tire Cornering Stiffness Adaptive Correction Estimation

The aforementioned tire lateral force estimation simulation conditions (1 and 2) were adopted in the simulation, and an adaptive correction estimator for the designed tire cornering stiffness was simulated and verified.

(1) Simulation results for working condition 1 are presented in Figure 6 and Figure 7.

Figure 6 shows the tire slip angle estimated using Equation (29). From the simulation results, the error between the estimated front-tire slip angle and the actual value output by CarSim was approximately 0°, and the maximal error between the rear-tire slip angle and the actual output value was only approximately 0.5533°. Hence, the slight deviation in this part was due to neglecting the rear wheel angle in the calculation model formula; however, this small deviation had very little effect on the estimation system, which was within the acceptable range.

In Figure 7 and Figure 8c–e, the purpose was to verify the effectiveness of the tire cornering stiffness adaptive correction estimation scheme proposed in this article using the tire cornering stiffness correction estimation results. Therefore, after obtaining the corrected estimate of the tire cornering stiffness (as shown in Figure 7c and Figure 8e), a linear tire model was used to calculate the tire lateral force and to multiply the corrected estimate of the tire cornering stiffness and the tire slip angle (calculated from Equation (29)) to obtain an estimate of the tire lateral force, as shown in Figure 7a,b and Figure 8c,d, and compare it with the actual output value of Carsim’s tire lateral force to verify the validity of the tire cornering stiffness correction estimate. The smaller the difference between the calculated tire lateral force estimate and the actual value, the better the corrected estimation result for the tire cornering stiffness.

The estimation results for the tire cornering stiffness under the complex working conditions of left and right turns are shown in Figure 7c. The product of the final tire cornering stiffness estimated through real-time correction and the tire sideslip angle is expressed as the tire lateral force estimation value. This was used to verify the validity and accuracy of the estimation results. The comparison results with the actual output values of CarSim are presented in Figure 7a,b. It is apparent that the maximum errors with the front- and rear-tire lateral forces correspond to 754.3698 and 430.5427 N. Further, from the simulation results, the fluctuation changes in the tire cornering stiffness estimation are consistent with the tire switching between the linear and nonlinear zones. Although the tire repeatedly switches between the linear and nonlinear zones under this working condition, the estimated results and response speed show excellent performance. Throughout the entire estimation process, the estimated tire lateral force based on the tire cornering stiffness correction approximation remained close to the true value and within a small error range.

(2) To test the robustness of the estimator with respect to various complex working conditions, working condition 2 was regarded for the simulation test verification.

Under the working conditions of low adhesion and double lane shifting at medium and high speeds, the simulation results in Figure 8a,b show that the error between the values estimated from the front- and rear-tire slip angle expressions according to Equation (29) and the actual value output by CarSim is almost 0°. Moreover, from Figure 8c–e, the maximum error between the estimated front-tire lateral force based on the tire cornering stiffness correction estimate and the true value is approximately 276.2459 N, and, considering the rear tire force at approximately 6.2 s, the error suddenly increased to 928 N but dropped to 0 in an instant; the overall error between the two was close to 0 N at many instances. It can be further found from Figure 8a,b that the tire slip angles $\alpha_{f}$ and $\alpha_{r}$ of the front and rear wheels, respectively, mostly exceed 4–5°. In this condition, the tire cornering characteristics enter the nonlinear region. If a constant linear tire cornering stiffness is considered to calculate the tire lateral force, it will lead to a large error in the true value. The real-time correction estimation strategy for tire cornering stiffness proposed in this study is based on the tire lateral force accurately estimated in real time by the UKF. When the tire is in the nonlinear zone, the tire cornering stiffness can be corrected and compensated to achieve a value as close as possible to the true value of the tire lateral force. This further improves the controller’s control performance and its universal applicability to road adhesion conditions.

In summary, the simulation verification and analysis of the above working conditions show that the tire lateral force estimation based on the UKF and the corresponding adaptive correction strategy for the tire cornering stiffness can provide accurate estimation of the tire cornering stiffness parameter value in real time with good adaptability to complex working conditions.

## 4. Design of Improved Adaptive Model Predictive Control for Trajectory Tracking

### 4.1. Design of MPC for Trajectory Tracking

Tracking using MPC is performed based on a path-tracking error dynamics model. Combined with the established error dynamics model, $\dot{X}=A_{1}X+B_{1}u+C_{1}{\dot{\theta }}_{r}$, the design of the MPC controller is as follows:

(1) Discretize the tracking error dynamics model.

After discretization processing, the following discretization formula is obtained:(33)X(k+1)=aX(k)+bu(k)+d
where a is the discretization matrix obtained using the mid-point Euler discretization method, $a={\left(I-\frac{A_{1}T}{2}\right)}^{-1}(I+\frac{A_{1}T}{2})$, T is the system discrete sampling time, $b=B_{1}T$, and $d=C_{1}{\dot{\theta }}_{r}T$. Therefore, the linear discretized error dynamics model after discretization is expressed as follows:(34)$$\left\{\begin{array}{l} X(k+1)=aX(k)+bu(k)+d\\ Y(k+1)=C_{2}X(k+1) \end{array}\right.$$
where Y(k+1) is the output equation and $C_{2}$ is the unit matrix. Because the above model does not include an increment in the control quantity, a discontinuity may be caused in the solved control quantity, reducing the smoothness of the vehicle. Therefore, the new state vector was constructed so that the new form contains constraints on the control increments.

Combining $\xi (k)={\left[X(k),u(k-1)\right]}^{T}$ with Equation (34), a new form of the prediction equation is obtained:(35)$$\left\{\begin{array}{l} \xi (k+1)=A\xi (k)+B∆u(k)+D\\ \eta (k+1)=C\xi (k+1) \end{array}\right.$$
where $A=\left[\begin{array}{cc} a & b\\ O & I \end{array}\right],\;B=\left[\begin{array}{c} b\\ I \end{array}\right],\;D=\left[\begin{array}{c} d\\ O \end{array}\right],\;\mathrm{and}\;C=[C_{2},O]$.

(2) The output in the prediction time domain is calculated according to the following prediction model.
(36)$$\left\{\begin{array}{l} \xi (k+1)=A\xi (k)+B∆u(k)+D\\ \xi (k+2)=A^{2}\xi (k)+AB∆u(k)+B∆u(k+1)+AD+D\\ \xi (k+3)=A^{3}\xi (k)+A^{2}B∆u(k)+AB∆u(k+1)+B∆u(k+2)+A^{2}D+AD+D\\ \;\;\;\;\;\;\;\;\;\vdots \\ \xi (k+N_{p})=A^{N_{p}}\xi (k)+A^{N_{p}-1}B∆u(k)+\cdots +B∆u(k+N_{c}-1)+A^{N_{p}-1}D+A^{N_{p}-2}D+\cdots +D \end{array}\right.$$
where $N_{p}$ and $N_{c}$ represent the prediction and control time domains, respectively, and the control increment Δu(k)=u(k)−u(k−1). In the above equations, let $Y(k)=[\eta (k+1),\eta (k+2),\cdots ,\eta (k+N_{p})]$; then:(37)$$Y(k)=\psi_{k}\xi (k)+\theta_{k}∆U(k)+\Gamma_{k}\Phi (k)$$
where $\psi_{k}={\left[CA,CA^{2},\cdots ,CA^{N_{p}}\right]}^{T}$,
$$\theta_{k}=\left[\begin{array}{cccc} CB & O & \cdots & O\\ CAB & CB & & \vdots \\ \vdots & \vdots & \ddots & O\\ CA^{N_{p}-1}B & CA^{N_{p}-2}B & \cdots & CA^{N_{p}-N_{c}}B \end{array}\right],\;\Gamma (k)=\left[\begin{array}{cccc} C & O & \cdots & O\\ CA & C & & \vdots \\ \vdots & \vdots & \ddots & O\\ CA^{N_{p}-1} & CA^{N_{p}-2} & \cdots & C \end{array}\right],\;\mathrm{and}\;\Phi (k)=\left[\begin{array}{c} D\\ D\\ \vdots \\ D \end{array}\right],\;∆U(k)=\left[\begin{array}{cccc} ∆u(k) & ∆u(k+1) & \cdots & ∆u(k+N_{c}-1) \end{array}\right].$$

(3) Design objective functions and transformed solutions for quadratic problems.

Considering the state tracking error and control increment as the optimization objective, the relaxation factor ε is introduced for the case in which the optimal solution cannot be found in the specified time. The following equation was designed for the objective function.
(38)$$J={{\sum_{i=1}^{N_{p}}\left\|\eta (k+i)-\eta_{ref}(k+i)\right\|}^{2}}_{Q}+{{\sum_{i=0}^{N_{c}-1}\left\|∆u(k+i)\right\|}^{2}}_{R}+\rho \varepsilon^{2}$$

This is converted into matrix form as follows:(39)$$J=Y^{T}\bar{Q}Y+∆U^{T}\bar{R}∆U+\rho \varepsilon^{2}$$
where ρ is the weight coefficient of the relaxation factor (ε). Combining Equations (37) and (39), after integration and simplification, Equation (39) can be transformed into a standard quadratic programming problem:(40)$$\min J=\frac{1}{2}X^{T}HX+f^{T}X$$
where $H=2\left[\begin{array}{cc} \theta^{T}\bar{Q}\theta +\bar{R} & 0\\ 0 & \rho \end{array}\right],\;f^{T}=\left[\begin{array}{cc} 2E^{T}\bar{Q}\theta & 0 \end{array}\right]$, and E=ψξ+ΓΦ.

Regarding the actual control driving conditions, the following constraints were set:(41)$$\left\{\begin{array}{l} ∆U_{\min}\leq ∆U(t+i|t)\leq ∆U_{\max}\\ U_{\min}\leq U(t+i|t)\leq U_{\max} \end{array}\right.$$

In addition, considering the control stability during vehicle driving, constraints such as the center-of-mass sideslip angle were added.
(42)$$\left\{\begin{array}{l} \alpha_{\min}\leq \alpha \leq \alpha_{\max}\\ \beta_{\min}\leq \beta \leq \beta_{\max} \end{array}\right.$$

### 4.2. Design of Dynamic Prediction Time-Domain Adaptive Model

In the MPC, the selection of the prediction time domain ($N_{p}$) has an important impact on the control effect of the system. If $N_{p}$ is small, the output will produce a large jitter; on the contrary, the system state error will be large, affecting the control accuracy. During the driving process of the vehicle, changes in its speed as well as road adhesion conditions (μ) significantly impact the vehicle trajectory tracking effect and vehicle stability [36]. It is thus necessary to analyze the impact of varying $N_{p}$ on the control effect under different driving speeds (v and μ).

#### 4.2.1. Calculation Method for Weight of Tracking Control Effect Indicators

The analytic hierarchy process (AHP) was employed to determine the weight of each of the evaluation indicators (i.e., lateral position error, heading angle error, and front wheel angle) of the tracking control effect. The specific process is described below.

Step 1: Construct judgment matrix *A* based on n indicators of the evaluation object. The indicator set of the evaluation object is $\left\{a_{1},a_{2},\cdots ,a_{n}\right\}$, using the 1–9 scaling method. The *i*-th and *j*-th indicators are compared according to their importance to obtain a judgment matrix $A={\left(a_{ij}\right)}_{n\times n}$.

Step 2: Choose an appropriate method to calculate the weight of each indicator. In this study, a general weight formula was used to calculate the indicator weight. The weight ($W_{i}$) of the *i*-th indicator can then be obtained using the following method:

The elements of each row of judgment matrix *A* are multiplied and squared n times to obtain ${W^{*}}_{i}$ and then the weights $W_{i}$:(43)$$\left\{\begin{array}{l} W_{i}^{*}=\sqrt[{n}]{\prod_{j=1}^{n}a_{ij}},\;i=1,2,\cdots n\\ W_{i}=W_{i}^{*}/\sum_{i=1}^{n}W_{i}^{*},\;i=1,2,\cdots n \end{array}\right.$$

Step 3: Check consistency. First, we summed the elements of each column of matrix *A* as:(44)$$S_{j}=\sum_{i=1}^{n}a_{ij},\left(j=1,2,\cdots ,n\right)$$

Then, the maximum eigenvalue ($\lambda_{\max}$) of matrix *A* was determined using the following equation:(45)$$\lambda_{\max}=\sum_{i=1}^{n}W_{i}S_{i}$$

When $\lambda_{\max}$ is greater than ${\lambda^{\prime }}_{\max}$, as given in Table 2, *A* cannot pass the consistency test, and the value of $a_{ij}$ should be adjusted to recalculate $\lambda_{\max}$ until $\lambda_{\max}$ is smaller than ${\lambda^{\prime }}_{\max}$.

Completing calculations according to the above steps, the weight value $W=(W_{1},W_{2},\cdots ,W_{n})$ is finally obtained for each indicator.

After obtaining each evaluation index value and the corresponding weight values, the comprehensive evaluation index value (*B*) of the trajectory-tracking control effect was calculated using the following equation:(46)$$B=(e_{d},e_{\psi },\delta_{f})\cdot W^{T}$$
where $e_{d},\;e_{\psi }$, and $\delta_{f}$ are the lateral error, the heading angle error, and the front-wheel turning angle index values, respectively.

#### 4.2.2. Design of Dynamic Prediction Time-Domain Adaptive Model

In this study, the following simulation test environment was set up: under the joint simulation platform of CarSim and Simulink, the vehicle speed was set as 30, 40 km/h, etc., at equal intervals of 10 km/h, up to 100 km/h; the corresponding road adhesion coefficients were 0.4, 0.5, 0.65, 0.8, 0.9, 0.95, and 1.0. Tracking control can be performed in CarSim with vehicles under the abovementioned driving conditions and a double-lane-shift path. Joint simulation experiments were also conducted considering different prediction time domains ($N_{p}$). Regarding the working conditions with a vehicle speed of 60 km/h and a road adhesion coefficient of 0.5, the control effect for varying $N_{p}$ is depicted in Figure 9. The three factors of lateral position error, heading angle error, and front-wheel rotation angle during path tracking (to ensure that there was no obvious strong jitter in the change in the front wheel angle) were considered as control effect indicators.

In Figure 9b–d, the lateral position error is minimized overall when the prediction time domain is 26°; however, the heading angle deviation is not minimized. Meanwhile, the corresponding front wheel angle appears to vibrate after approximately 5 s, which affects the comfort. However, when the prediction time domain is 30, although the lateral error is slightly larger than when the prediction time domain is 26, these three factors are significantly better than when the prediction time domain is 26, and the front wheel angle changes smoothly without jitter. Therefore, the final result is based on the AHP, and the optimal prediction time-domain value under this working condition is 30.

Similarly, based on the above analysis method, each driving condition was compared and analyzed separately to select the best prediction time domain for different road adhesion conditions and vehicle speeds. The results are summarized in Table 3.

The best prediction time domains under different driving conditions, as listed in Table 3, were used to fit and solve the relationship between μ, v, and $N_{p}$ using a three-dimensional surface function. Finally, the $N_{p}$ value obtained by the interpolation fitting solution was rounded to realize the final prediction time domains, as depicted in Figure 10. In addition, the dynamic prediction time-domain adaptive model was called the AD model for short.

### 4.3. Improved AMPC Strategy for Trajectory Tracking

Combining the designed adaptive correction estimator for tire cornering stiffness and the dynamic prediction time-domain adaptive model, an improved AMPC strategy for trajectory tracking was established. Under different working conditions, the tire lateral force calculation error was considered to correct and estimate tire cornering stiffness in real time to obtain the AMPC, ensuring the stability of tracking control and improving the controller’s performance; meanwhile, the AMPC was further improved to enable dynamic selection of the best prediction time domain under different road adhesion conditions and vehicle speeds, further ensuring the accuracy and stability of tracking control. Finally, the improved AMPC method for trajectory tracking was achieved, as shown in Figure 11.

## 5. Simulation Test and Verification

With the CarSim and MATLAB/Simulink joint simulation platform, the improved AMPC proposed herein, which combines the adaptive correction estimation for tire cornering stiffness and the dynamic prediction time-domain adaptive model, was simulated and verified.

For most normal working conditions or complex and severe working conditions, such as low-adhesion single-lane-shift working conditions (slightly higher vehicle speed) or low-adhesion double-lane-shift working conditions, some studies in the relevant literature (such as references [37,38,39,40,41]) have shown that path tracking controllers based on MPC have better path-tracking control effects than the LQR controller and the pure tracking controller. Therefore, this article will no longer jointly compare the tracking control performance of the LQR path tracking controller and the pure tracking controller with that of the ordinary MPC controller and the improved AMPC controller proposed in this article. Only the improved AMPC controller mentioned in this article and the ordinary MPC controller will be simulated, compared, and analyzed to conduct a simulation comparison and verification of the proposed controller.

Generally speaking, the lateral acceleration of ordinary passenger cars under normal driving conditions is usually between 0.8 g and 1.2 g (1 g is 9.8 m/s^2^); under ideal circumstances, a vehicle will begin to slide when the lateral acceleration reaches the acceleration provided by the maximum static friction between the tire and the road. This critical lateral acceleration can be estimated by the following formula:(47)$$a_{y\;crit}=\mu g$$
where $a_{y\;crit}$ is the critical lateral acceleration of the car when it slips, μ is the road adhesion coefficient, and g is the gravity acceleration, which is 9.8 m/s^2^.

The results of Bosch’s research on vehicle stability show [42,43] that on roads with good adhesion, such as dry asphalt pavement, the center-of-mass sideslip angle limit for a vehicle to drive stably can reach $\pm 12^{{}^{\circ}}$; while on roads with low adhesion, such as icy and snowy roads, the limit value is approximately $\pm 2^{{}^{\circ}}$.

In summary, when conducting simulation experiments in this article, when the vehicle under the control of the controller was tracking the path, the vehicle center-of-mass sideslip angle (β) and the vehicle lateral acceleration ($a_{y}$) were used to initially judge the driving stability of the vehicle when tracking. In addition, since the focus of this study is vehicle path tracking control under improved AMPC control, the verification of the simulation results mainly focused on the tracking accuracy of path tracking and at the same time ensured the stability of tracking driving. Therefore, we only made a superficial preliminary judgment on vehicle driving stability based on β and $a_{y}$ and did not conduct more in-depth research and analysis on vehicle stability.

### 5.1. Medium–High Speed and Low-Adhesion Single-Lane-Shift Working Conditions

First, the control performance of the improved AMPC was verified based on a typical single-lane-shift condition, with the vehicle speed set to 70 km/h and a roadway adhesion coefficient of 0.4. As shown in Figure 12, the path-tracking control effect based on the improved AMPC is better than that of the traditional MPC, and the tracking accuracy is significantly improved. Compared with the ordinary MPC, the proposed improved AMPC achieves a maximum reduction in the path-tracking lateral position error of 0.3152 m (time of 4.652 s), and the heading angle error is significantly reduced in the time range of 3.967–4.588 s. At approximately 4.146 s, the error decreased by 2.3902°. It is apparent from Figure 12e that the front-tire sideslip angle ($\alpha_{f}$) began to exceed 4° at 3.436 s and even reached a maximum of approximately 7.4°. At this time, the tire was already operating in a nonlinear zone. There is a big difference between the tire lateral force calculated based on the linear cornering stiffness ($C_{\alpha }$) and the real value, so it is difficult for the controller to solve the optimal front-wheel angle control quantity. Through the adaptive correction estimation of the tire cornering stiffness with the improved AMPC, the calculated tire force can be as close to the real value as possible, and the vehicle can be controlled to better track the reference path. As shown in Figure 12d, the front wheel angle changes more gently based on the improved AMPC than with the ordinary MPC, and the jitter is reduced; hence, the driving is more stable and comfortable when following the path.

In Figure 12g, it can be seen from the above that on low-adhesion road surfaces, the upper and lower limits of the vehicle’s center-of-mass slip angle for stable driving are $\pm 2^{{}^{\circ}}$ and that the vehicle’s center-of-mass slip angles under MPC control and improved AMPC control are generally consistent; the maximum value is about 1.3°, which does not exceed the limit of the center-of-mass sideslip angle. The center-of-mass sideslip angle based on the improved AMPC control is slightly larger than that under MPC control at certain times. This is mainly due to the fact that the front wheel angle under the improved AMPC control is slightly larger than that under MPC control in Figure 12d. In Figure 12h, it can be seen from the above that since the road adhesion coefficient in this working condition is 0.4, the critical lateral acceleration of the vehicle to slide is 0.4 g. It can be seen that the lateral acceleration of the vehicle under the control of the two controllers does not exceed the acceleration limit, but the lateral acceleration based on the improved AMPC control is slightly increased compared to the MPC control. The main reason is the same as that for the change in the center-of-mass sideslip angle in Figure 12g, namely, that the front wheel angle under the improved AMPC control is slightly larger than that under the MPC control in Figure 12d. Overall, although the improved AMPC control does not significantly reduce the vehicle center-of-mass sideslip angle and lateral acceleration compared to the MPC control, the maximum values are maintained within a stable range, and the vehicle will not suffer from unstable situations such as sideslip, indicating that under this severe working condition, the improved AMPC controller proposed in this article not only improves the path tracking accuracy, but also ensures the driving stability of the vehicle.

According to the comparative analysis, the proposed improved AMPC has higher control performance than the traditional MPC in the single-lane-shift condition of low adhesion and medium–high speed.

### 5.2. Medium–High Speed and Low-Adhesion Double-Lane-Shift Working Conditions

Based on the typical double-lane-shift condition, to verify the control performance of the improved AMPC and its adaptability to the vehicle driving state, the vehicle speed was set to 60 km/h and the road adhesion coefficient was 0.4.

From Figure 13a–c, it is apparent that the path tracking control based on the improved AMPC controller is better than that based on the ordinary MPC controller and that the tracking accuracy is greatly improved. The maximum lateral position error under the traditional MPC was 0.6574 m. Meanwhile, the proposed improved AMPC has a maximum lateral position error for path tracking of 0.5623 m, a reduction of 14.47%. Further, the maximum values of the heading angle deviation under the control of the improved AMPC and the ordinary MPC are the same, but in the time range of 4.412–5.314 s, the heading angle deviation of the improved AMPC is significantly lower than that of the ordinary MPC, and the maximum decrease is 0.8378°. At 5.604 s, compared with the ordinary MPC controller, the heading angle deviation was reduced by a maximum of 1.0947°.

In addition, as shown in Figure 13d, based on the improved AMPC compared with the ordinary MPC, the front wheel angle reduces at approximately 3–4 s, and from 4.6 s onwards, the front wheel angle also decreases more gently, although there is a slight amplitude value oscillation when finally approaching the value of 0, but, in general, this does not affect the control performance, and the front wheel angle is clearly improved. In Figure 13e, the overall difference in the center-of-mass sideslip angle based on the improved AMPC compared with that of the normal MPC is not significant, but the center-of-mass sideslip angle slightly decreased at approximately 5.218 s.

In Figure 13e, since this working condition is a low-adhesion road condition with an adhesion coefficient of 0.4, it can be seen from the above that the upper and lower limits of the center-of-mass sideslip angle of the vehicle for stable driving are $\pm 2^{{}^{\circ}}$. The vehicle center-of-mass sideslip angles under MPC control and improved AMPC control are generally consistent, and the maximum center-of-mass sideslip angle under improved AMPC control is about 1°, which is lower than the upper limit. In Figure 13f, the lateral acceleration of the vehicle under improved AMPC control and MPC control is also roughly the same, and the maximum value does not exceed the critical acceleration of 0.4 g for vehicle sideslip. Comprehensive analysis of Figure 13e,f shows that under this severe working condition, the vehicle will not suffer dangerous situations such as sideslip under the action of the proposed controller, which ensures the stability of the vehicle while driving to a certain extent. To sum up, the proposed improved AMPC controller not only improves the path tracking accuracy, but also ensures the driving stability of the vehicle.

### 5.3. High-Speed and High-Adhesion Double-Lane-Shift Working Conditions

To further verify the adaptability of the proposed improved AMPC to road adhesion conditions and control performance, tests considering the double-lane-shift condition were continued, with the vehicle speed set to 80 km/h and a road adhesion coefficient of 0.9. A comparison of the simulation results is presented in Figure 14.

From Figure 14a–c, the path tracking control based on the improved AMPC again appears better than that of the ordinary MPC, leading to improved tracking accuracy and control performance. For specific analysis, the maximum lateral position error under normal MPC is 0.5578 m, while that under the improved AMPC is 0.4746 m, a reduction of 14.92%. Compared with the traditional MPC, the heading angle deviation under the improved AMPC also partially reduces at 2.203–2.635 s. The overall control effect of the front wheel angle is similar. Further considering this beside Figure 14e, the vehicle center-of-mass sideslip angle is also partially reduced, and the changing range of the respective value also reflects that the vehicle is in a normal working condition.

In Figure 14e, since the working condition is a high-adhesion road condition with an adhesion coefficient of 0.9, it can be seen from the above that the upper and lower limits of the center-of-mass sideslip angle of the vehicle that can drive stably are $\pm 12^{{}^{\circ}}$ and that the maximum sideslip angle of the vehicle’s center of mass controlled by the improved AMPC is 2°, which is far lower than the upper limit. In Figure 14f, the lateral acceleration of the vehicle based on the improved AMPC control is roughly the same as that under MPC control, and the maximum value is about 8°, which is lower than the critical lateral acceleration of 0.9 g (8.82 m/s^2^) when the vehicle sideslips. Similarly, the comprehensive analysis of Figure 14e,f shows that under this working condition, the vehicle will not suffer dangerous situations such as sideslip under the action of the proposed controller, which ensures the stability of the vehicle while driving to a certain extent.

It can be concluded from this working condition and the simulation results presented in Section 5.2 that, based on the adaptive correction estimation of the tire cornering stiffness in the improved AMPC, the adaptability of the MPC to road adhesion conditions can be improved while guaranteeing vehicle path tracking accuracy and stability.

### 5.4. Docking Road Conditions

To further verify the improvement effect of the proposed dynamic prediction time-domain adaptive model on the improved AMPC, the simulations were verified via tracking the double-lane-shift trajectory at the vehicle speed of 50 km/h on a docking road surface.

Based on the typical double-lane-shift conditions to verify the control performance of the improved AMPC and the adaptability to the vehicle driving state, the vehicle speed was set to 50 km/h on a docking pavement (road adhesion coefficient (μ) of 0.85–0.4) and the road adhesion coefficient was 0.85 for a longitudinal displacement of 0–53 m and 0.4 for a displacement of 53–120 m. The docking road conditions are depicted in Figure 15, and a comparison of the simulation results is provided in Figure 16.

The MPC that incorporates the proposed dynamic prediction time-domain adaptive model is referred to as the improved MPC, and the MPC that combines the dynamic prediction time-domain adaptive model and tire cornering stiffness adaptive correction model is referred to as the improved AMPC. As can be seen in Figure 16, when the vehicle speed is 50 km/h on the docking road and the prediction time-domain steps are selected as 12, 17, and 22, the controller cannot track the desired path. Further, when $N_{p}$ was set to 27, 32, 37, and 42, although the controller could track the reference path, the tracking error was large. When the prediction time domain is 17 and 22, it can be seen that within 0–53 m, due to good road adhesion conditions, the controller can track the desired path. However, when the displacement reaches 53–120 m under low-adhesion road conditions, the tracking error is large owing to the controller’s insufficient prediction of the external environment: the desired path can thus not be tracked, and the control effect is poor.

When the prediction time domain is selected as the step size obtained under the AD model, the controller can track the entire variable attachment section of the reference path, indicating that the designed dynamic prediction time-domain adaptive model is correct. At 0–53 m of the high-adhesion road section, the AD model calculates a smaller step size of 19 for the prediction time domain, which improves the calculation efficiency. At 53–120 m, the AD model calculates an $N_{p}$ of 38, which can accurately predict the future state of the vehicle as well as improve the tracking accuracy and driving stability. The changes in the values calculated using the dynamic prediction time-domain model are exhibited in Figure 16f.

In Figure 16, for the docking road condition with the road adhesion coefficient from 0.85–0.4, it can be found from Figure 16e that the vehicle center-of-mass sideslip angle under the improved AMPC control is reduced overall compared to the MPC control and that the center of mass the maximum value of the center-of-mass sideslip angle is less than 1°, which is far lower than the upper limit of the center-of-mass sideslip angle when the vehicle is driving stably under this adhesion coefficient condition. In Figure 16g, when the vehicle is driving on a road where the road adhesion coefficient changes, the critical acceleration without sideslip is 0.85–0.4 g and the maximum value of vehicle lateral acceleration based on improved AMPC control is 3.704°, which does not even exceed 0.4 g (3.920°), and the lateral acceleration changes relatively smoothly. The analysis of Figure 16e,g shows that under the action of the proposed controller, the vehicle will not suffer dangerous situations such as sideslip, and the stability of the vehicle while driving is guaranteed to a certain extent.

In addition, from Figure 16b,c, on the basis of adding a dynamic prediction time-domain adaptive model (i.e., improved MPC) combined with the tire cornering stiffness adaptive correction (i.e., improved AMPC), the lateral position error and heading angle error of the vehicle tracking reference path are further reduced. Combined with Figure 16e, the stability of the vehicle driving tracking is also ensured.

## 6. Conclusions

(1) A UKF was adopted to accurately estimate the tire lateral force in real time. With the vehicle body dynamics model and observed quantities, i.e., longitudinal vehicle speed and lateral and longitudinal acceleration, the prediction and measurement equations were modeled and then combined with the UKF to complete estimation of the tire lateral force. The simulated working conditions were verified and show that the proposed method can estimate the lateral tire force accurately and has excellent applicability in complex working conditions.

(2) Based on the real-time estimation of the tire lateral force, an adaptive tire cornering stiffness correction strategy was proposed. The linear tire lateral force calculated using a constant tire cornering stiffness has a certain error with respect to the real value, particularly under complex road conditions, and this error is large. With the proposed correction strategy, the tire lateral force calculated by the controller is as close as possible to the real value, as the tire cornering stiffness is compensated for in real time, improving the tracking accuracy and driving stability.

(3) During the path tracking of the vehicle, changes in vehicle speed and road adhesion conditions significantly impact the vehicle trajectory tracking effect and vehicle stability. Therefore, a dynamic prediction time-domain adaptive model was introduced into the MPC (i.e., AMPC) that includes adaptive correction estimation for the tire cornering stiffness. The prediction time-domain value is dynamically obtained according to the road adhesion condition and vehicle speed, so the trajectory is tracked more accurately, as demonstrated by the corresponding simulation verification.

Overall, compared with the traditional MPC, the proposed improved AMPC has enhanced trajectory tracking accuracy and driving stability under different road adhesions, is more adaptive to different road conditions, can handle control instability caused by sudden changes in road adhesion, and exhibits improved tracking accuracy. Thus, the proposed AMPC method is of great significance for improving the adaptability and robustness of intelligent vehicle tracking control systems.

## Figures and Tables

**Figure 1 sensors-24-02316-f001:**
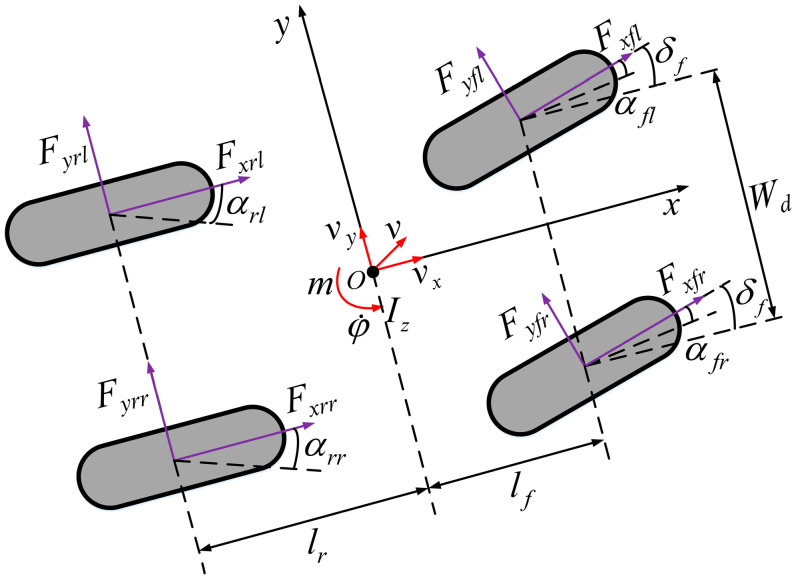
Four-wheel body model.

**Figure 2 sensors-24-02316-f002:**
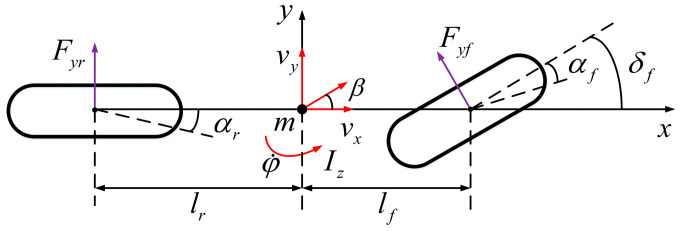
Two-degrees-of-freedom dynamics model of vehicle.

**Figure 3 sensors-24-02316-f003:**
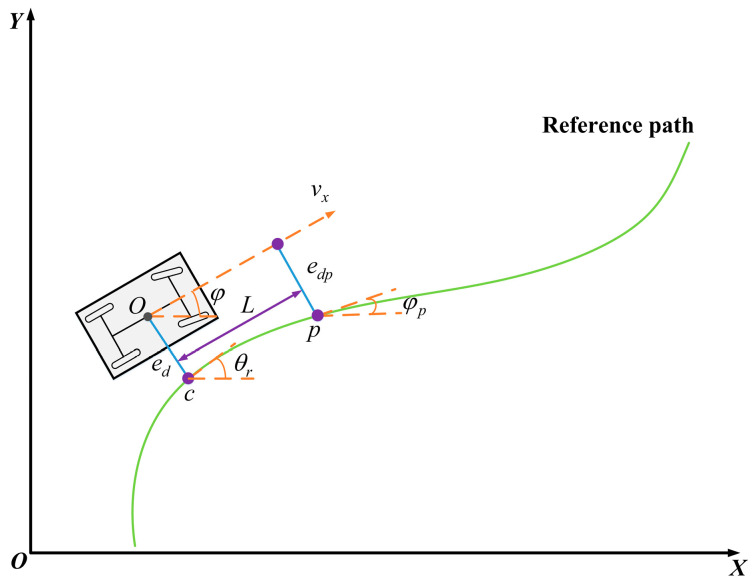
Vehicle–road error model.

**Figure 4 sensors-24-02316-f004:**
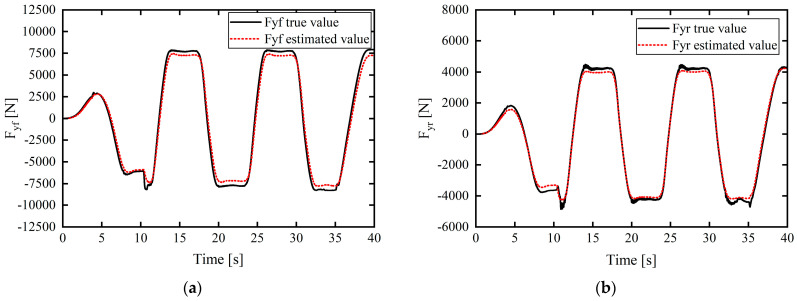
Tire lateral force estimation results under working condition 1: (**a**) front-wheel lateral force; (**b**) rear-wheel lateral force.

**Figure 5 sensors-24-02316-f005:**
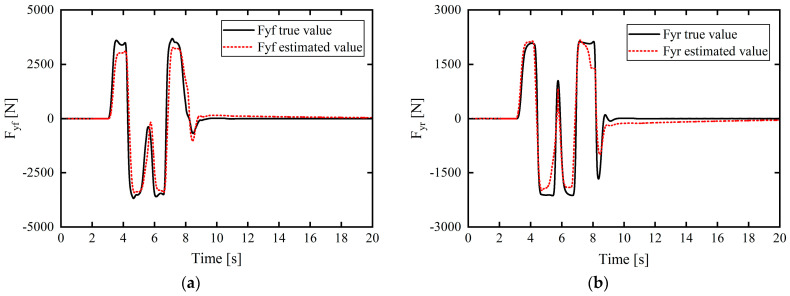
Tire lateral force estimation results under working condition 2: (**a**) front-wheel lateral force; (**b**) rear-wheel lateral force.

**Figure 6 sensors-24-02316-f006:**
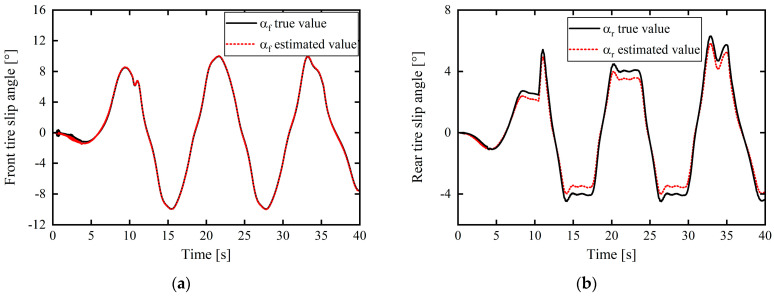
Tire slip angle estimation results: (**a**) front-tire slip angles; (**b**) rear-tire slip angles.

**Figure 7 sensors-24-02316-f007:**
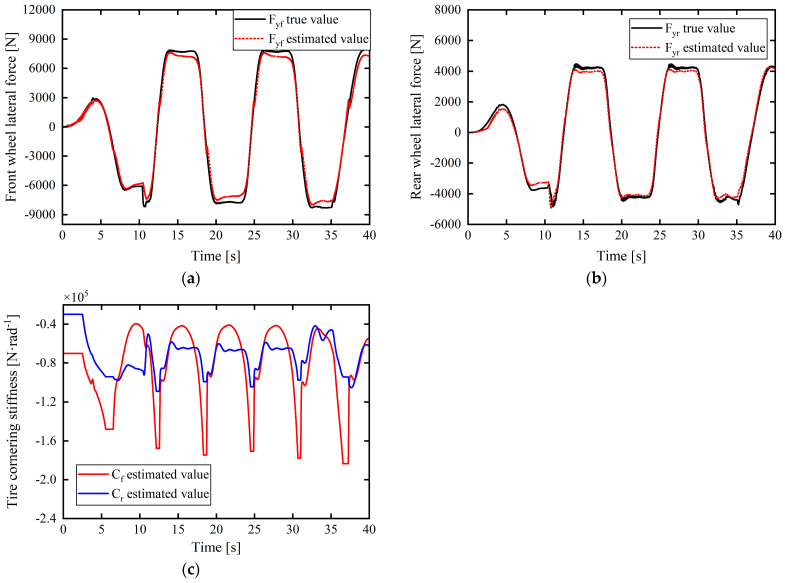
Real-time correction estimation and verification of front- and rear-tire cornering stiffnesses: (**a**) front-wheel lateral force; (**b**) rear-wheel lateral force; (**c**) front- and rear-tire cornering stiffnesses.

**Figure 8 sensors-24-02316-f008:**
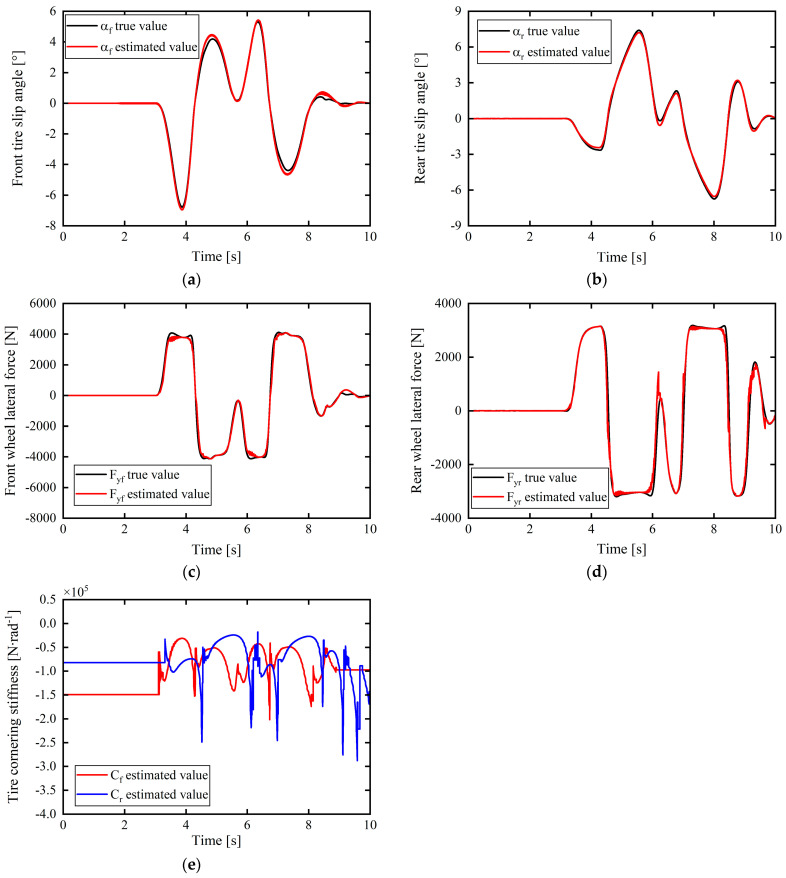
Estimation and verification of front- and rear-tire cornering stiffnesses: (**a**) front-tire slip angles; (**b**) rear-tire slip angles; (**c**) front-wheel lateral force; (**d**) rear-wheel lateral force; (**e**) front- and rear-tire cornering stiffnesses.

**Figure 9 sensors-24-02316-f009:**
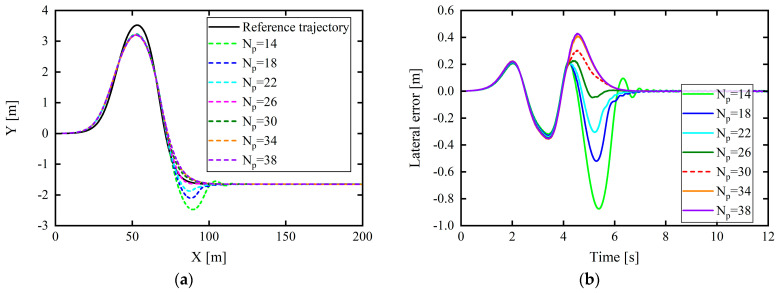
Control under different prediction time domains (*N_p_*) for vehicle speed of 60 km/h and road adhesion coefficient of 0.5: (**a**) tracking comparison of reference trajectories; (**b**) lateral position errors; (**c**) heading angle errors; (**d**) front wheel angle.

**Figure 10 sensors-24-02316-f010:**
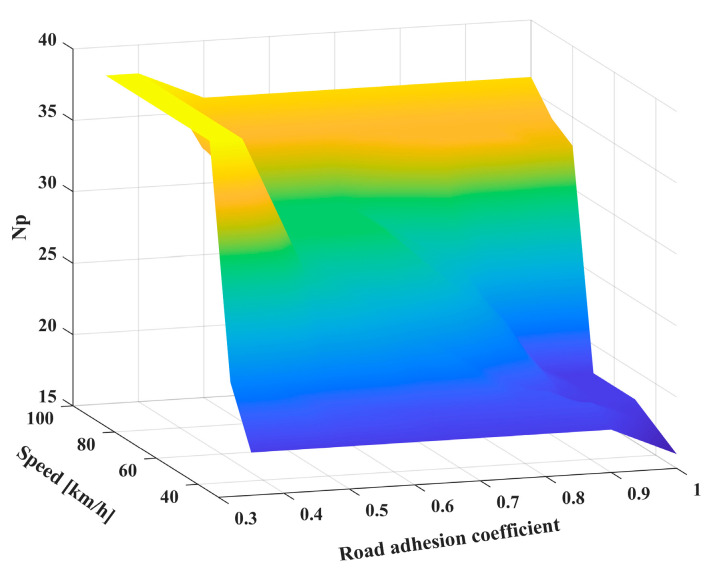
*N_p_* under three-dimensional surface function fitting.

**Figure 11 sensors-24-02316-f011:**
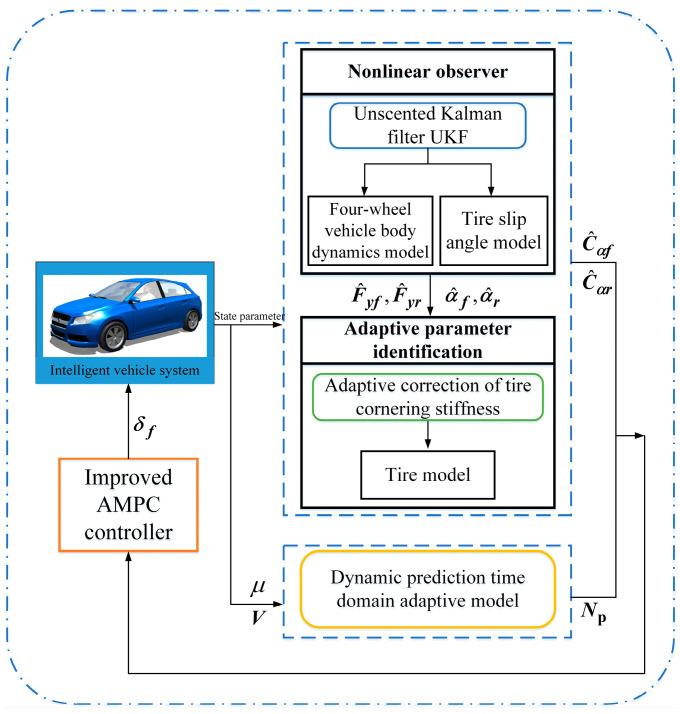
Block diagram of improved adaptive model predictive control (AMPC) strategy.

**Figure 12 sensors-24-02316-f012:**
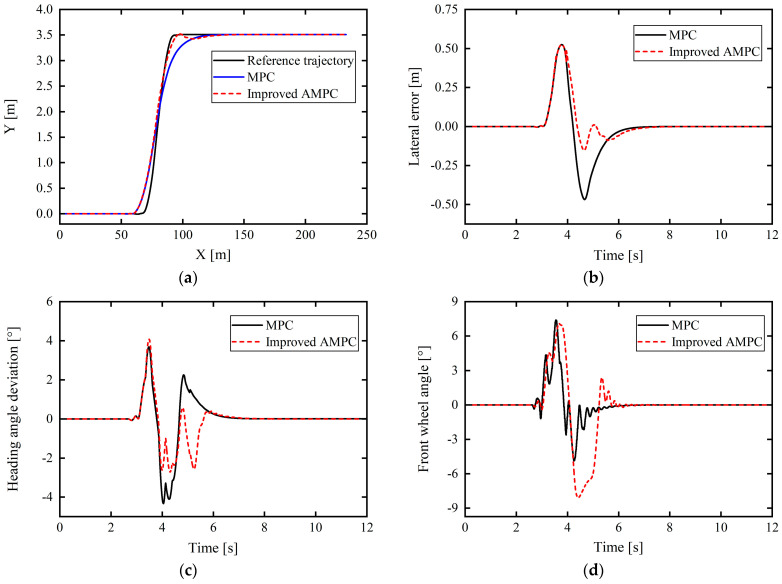
Comparison of control effects under medium–high speed and low-adhesion single-lane-shift working condition: (**a**) tracking comparison of reference trajectories; (**b**) lateral position errors; (**c**) heading angle errors; (**d**) front wheel angle; (**e**) front-tire slip angle (*α_f_*); (**f**) rear-tire slip angle (*α*)*_r_*; (**g**) vehicle center-of-mass sideslip angle; (**h**) lateral acceleration.

**Figure 13 sensors-24-02316-f013:**
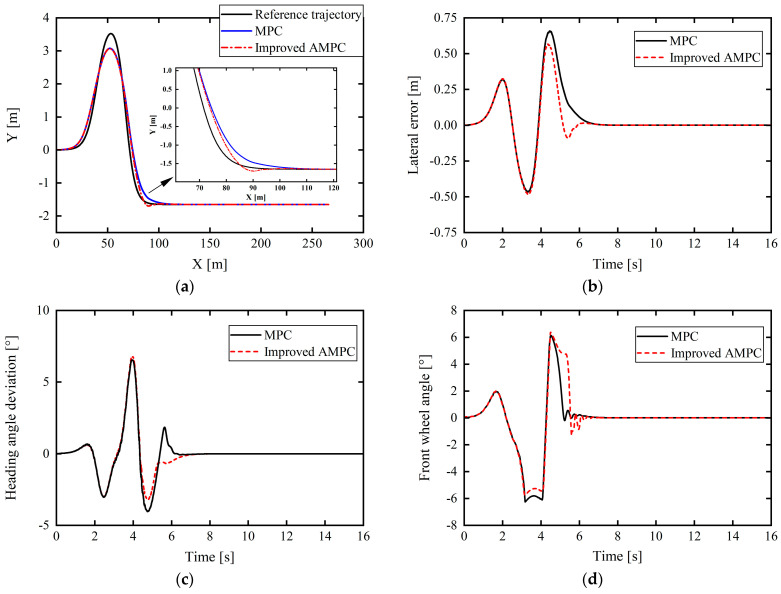
Comparison of control effects under medium–high speed and low-adhesion double-lane-shift working conditions: (**a**) tracking comparison of reference trajectories; (**b**) lateral position errors; (**c**) heading angle errors; (**d**) front wheel angle; (**e**) vehicle center-of-mass sideslip angle; (**f**) lateral acceleration.

**Figure 14 sensors-24-02316-f014:**
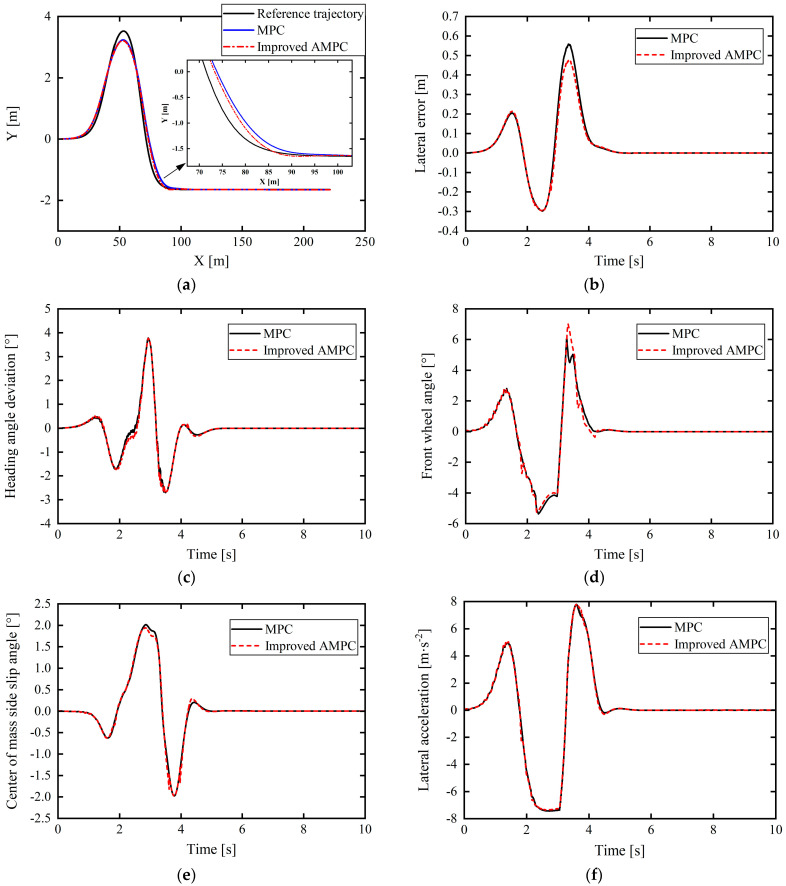
Comparison of control effects under high-speed and high-adhesion double-lane-shift working conditions: (**a**) tracking comparison of reference trajectories; (**b**) lateral position errors; (**c**) heading angle errors; (**d**) front wheel angle; (**e**) vehicle center-of-mass sideslip angle; (**f**) lateral acceleration.

**Figure 15 sensors-24-02316-f015:**
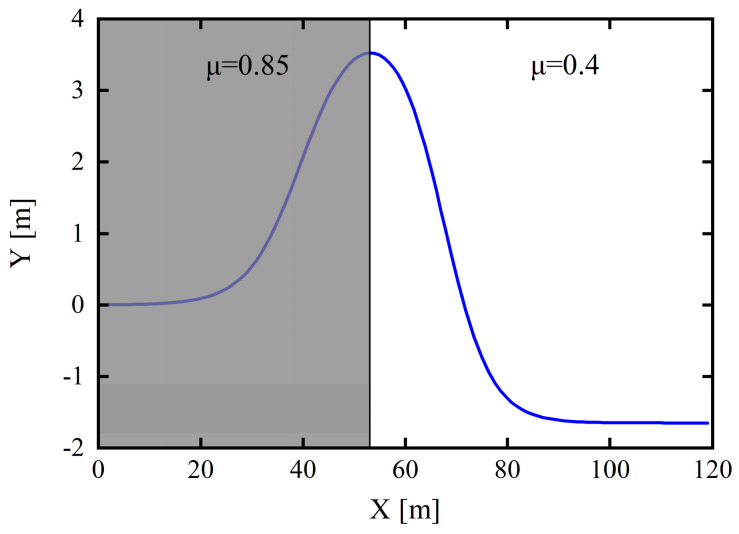
Expected trajectory under docking road surface.

**Figure 16 sensors-24-02316-f016:**
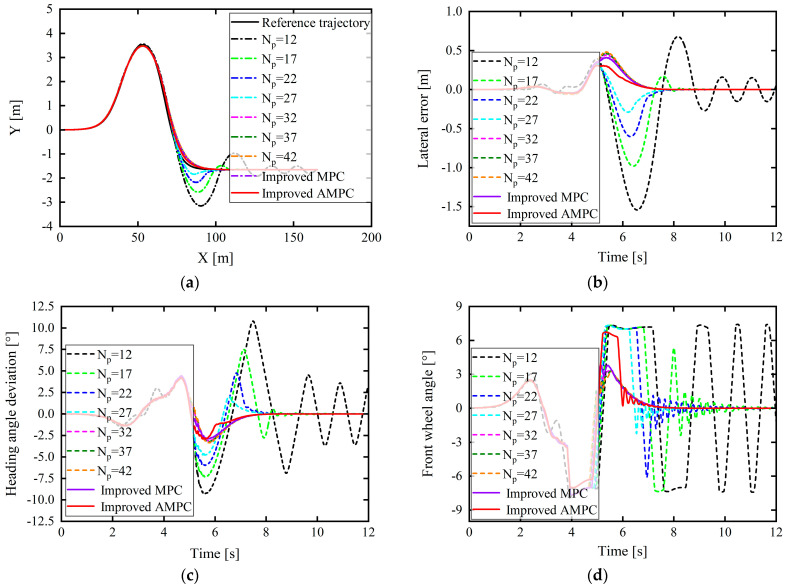
Contrast in control effects of docking road condition: (**a**) tracking comparison of reference trajectories; (**b**) lateral position errors; (**c**) heading angle errors; (**d**) front wheel angle; (**e**) vehicle center-of-mass sideslip angle; (**f**) change in *N_p_*; (**g**) lateral acceleration.

**Table 1 sensors-24-02316-t001:** Main vehicle parameters.

Vehicle Parameter	Value
Total vehicle mass (M/kg)	1412
Left and right wheelbase $\left(W_{d}/m\right)$	1.675
Distance from front axle to vehicle c.g. $\left(l_{f}/m\right)$	1.015
Distance from rear axle to vehicle c.g. $\left(l_{r}/m\right)$	1.895
Vehicle moment of inertia about *z*-axis $\left(I_{z}/\mathrm{kg}\cdot m^{2}\right)$	1536.7
Center-of-mass height (h/m)	0.54

**Table 2 sensors-24-02316-t002:** Corresponding critical eigenvalues of *n*-order matrix.

n	${\lambda^{\prime }}_{\max}$
3	3.116
4	4.27
5	5.45
6	6.62
7	7.79
8	8.99
9	10.16
10	11.34

**Table 3 sensors-24-02316-t003:** Best prediction time domains under different vehicle speeds and road adhesion conditions.

µ		V (km/h) 30	V 40	V 50	V 60	V 70	V 80	V 90	V 100
0.35	N_p_	18	22	38	38	38	38	38	38
0.4		18	22	38	38	38	38	38	38
0.5		18	20	28	30	30	30	34	36
0.65		18	19	24	30	30	30	34	36
0.8		18	19	20	24	26	34	34	36
0.9		18	19	18	19	19	34	34	36
0.95		17	18	18	18	18	33	34	36
1.0		16	17	18	17	17	33	34	36

## Data Availability

Data are contained within the article.

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
