# Peer review of "Research on Intelligent Vehicle Trajectory Tracking Control Based on Improved Adaptive MPC"

_sensors, 2024, doi:10.3390/s24072316_

Round 1

Reviewer 1 Report

Comments and Suggestions for Authors

The authors have proposed an adaptive MPC for four wheeled mobile robot with steering capability. The work is interesting and detailed. However, there are a few points that need clarifications:

1. MPC requires discretization and linearization of system dynamics, which invited the scope of modelling error. Authors need to improve the literature survey where such issues are avoided via continuous time adaptive control (see "Adaptive sliding mode control for autonomous vehicle platoon under unknown friction forces", "An output feedback based robust saturated controller design for pavement sweeping self-reconfigurable robot").

2. Authors have claimed to propose adaptive MPC. However, it is not clear where modelling uncertainty is taken.

3. Simulation results must be given in comparison with state of the art.

4. Closed-loop stability result is apparently missing. 

Author Response

I've modified it based on your suggestion. Please see the attachment for the reply content.

Reviewer 2 Report

Comments and Suggestions for Authors

The authors proposed a improved AMPC method for trajectory tracking and  the proposed AMPC technique were verified via co-simulation using CarSim and MATLAB/Simulink. The simulation results are adaptabile and robustness to the simulation vehicle tracking control systems, but there are some errors which should be discussed and modified.

1. In Eq. (14), the fourth equation  Fzfl should be modified wiht Fzfr. Please correct the mistake.

2.  In Fig. 4-8, how to obtain the true value of the model? They should be explained. If the comparison results are the actual output values of CarSim , it should be commented. How to obtain the estimated value? 

3. In the Joint simulation experiments, Np is conducted considering different prediction time domains with respect to different road adhesion conditions. 

How to determine the actual road adhesion coefficient in the actual experiments?

4. In the  comparison of control effects under medium-high speed and low-adhesion single-lane-shift working conditions, why does the proposed improved AMPC have larger front wheel angles in Figs. 12-14? 

5. In Fig. 16, please change the color of the improved MPC in Fig. (a) to obtain the consistency with the others. 

Author Response

(The authors gave the same response as above.)

Round 2

Reviewer 1 Report

Comments and Suggestions for Authors

No further comments